# SORL1 Polymorphisms in Mexican Patients with Alzheimer’s Disease

**DOI:** 10.3390/genes13040587

**Published:** 2022-03-25

**Authors:** Danira Toral-Rios, Elizabeth Ruiz-Sánchez, Nancy Lucero Martínez Rodríguez, Marlene Maury-Rosillo, Óscar Rosas-Carrasco, Fernando Becerril-Pérez, Francisco Mena-Barranco, Rosa Carvajal-García, Daniela Silva-Adaya, Yair Delgado-Namorado, Gerardo Ramos-Palacios, Carmen Sánchez-Torres, Victoria Campos-Peña

**Affiliations:** 1Department of Psychiatry, Washington University School of Medicine, St. Louis, MO 63110, USA; datr50@hotmail.com; 2Laboratorio de Neurotoxicología, Instituto Nacional de Neurología y Neurocirugía, Manuel Velasco Suárez, Ciudad de México 14269, Mexico; ruizruse@yahoo.com.mx; 3Unidad de Investigación Epidemiológica en Endocrinología y Nutrición, Hospital Infantil de México Federico Gómez, Ciudad de México 06720, Mexico; amr70@hotmail.com; 4Departamento de la Subdirección de Prevención y Protección a la Salud, Dirección Normativa de Salud, Instituto de Seguridad y Servicios Sociales de los Trabajadores del Estado (ISSSTE), Ciudad de México 14070, Mexico; marmmaury@gmail.com; 5Departamento de Salud, Universidad Iberoamericana, Ciudad de México 01219, Mexico; oscar_rosas_c@hotmail.com; 6Research Institute of Molecular Pathology (IMP), Vienna Biocenter (VBC), Campus-Vienna-BioCenter 1, 1030 Vienna, Austria; fernando.becerril@imp.ac.at; 7Hospital General ISSSTE, La Paz 23090, Mexico; fmenabarranco@yahoo.com.mx; 8Centro Geriátrico SINANK’AY, Jurica, Santiago de Querétaro 76100, Mexico; rosacarvajal@sinankay.net; 9Laboratorio Experimental de Enfermedades Neurodegenerativas, Instituto Nacional de Neurología y Neurocirugía, Ciudad de México 14269, Mexico; dan04siad@hotmail.com; 10Molecular Biology Laboratory, National Reference Center “Mexico’s Valley”, Salud Digna, Los Reyes, Tlal nepantla de Baz, Estado de Mexico 54075, Mexico; yaemde9@gmail.com; 11Department of Neurology and Neurosurgery, Montreal Neurological Institute, McGill University, Montréal, QC H3A 2B4, Canada; gerardora11@gmail.com; 12Departamento de Biomedicina Molecular, Centro de Investigación y de Estudios Avanzados del Instituto Politécnico Nacional, Ciudad de México 07360, Mexico; csanchez@cinvestav.mx

**Keywords:** AβPP processing, AβPP sorting, APOE genotype, sortilin 1, polymorphic variants

## Abstract

The present study evaluated the risk effect of 12 Single Nucleotide Polymorphisms in the SORL1 gene in the Mexican population using Late-Onset Alzheimer’s Disease (LOAD) and control subjects. Considering APOE as the strongest genetic risk factor for LOAD, we conducted interaction analyses between single nucleotide polymorphisms (SNPs) and the APOE genotype. Methods: Patients were interviewed during their scheduled visits at neurologic and geriatric clinics from different institutions. The LOAD diagnosis included neurological, geriatric, and psychiatric examinations, as well as the medical history and neuroimaging. Polymorphisms in *SORL1* were genotyped by real-time PCR in 156 subjects with LOAD and 221 controls. APOE genotype was determined in each study subject. Allelic, genotypic, and haplotypic frequencies were analyzed; an ancestry analysis was also performed. Results: The A/A genotype in rs1784933 might be associated with an increased LOAD risk. Two blocks with high degree linkage disequilibrium (LD) were identified. The first block composed by the genetic variants rs668387, rs689021 and rs641120 showed a positive interaction (mainly the rs689021) with rs1784933 polymorphism. Moreover, we found a significant association between the *APOE* ε4 allele carriers and the variant rs2070045 located in the second LD block. Conclusion: The rs1784933 polymorphism is associated with LOAD in Mexican patients. In addition, the presence of *APOE* ε4 allele and SORL1 variants could represent a genetic interaction effect that favors LOAD risk in the Mexican population. SNPs have been proposed as genetic markers associated with the development of LOAD that can support the clinical diagnosis. Future molecular studies could help understand sporadic Alzheimer’s Disease (AD) among the Mexican population, where currently there is a sub-estimate number in terms of disease frequency and incidence.

## 1. Introduction

The sortilin-related receptor (SORL1) is a member of the low-density lipoprotein-receptor family expressed in the human brain. The gene is located on 11q23.2-q24.2 and encodes a 250-kD protein [1]. Similarly to the apolipoprotein E (APOE), this type-1-membrane glycoprotein [2] can interact with the amyloid-β precursor-protein (AβPP), modulating its subcellular trafficking [1,3], and may influence the amyloid-β (Aβ) production [4,5]. The major genetic risk factor linked to Late-Onset Alzheimer’s Disease (LOAD) is the presence of one or two copies of the ε4 allele in the APOE gene, but this is neither a necessary nor sufficient condition for developing the disease. Genome–Wide association studies have revealed new candidate genes related to the disease, such as SORL1 [6,7,8,9,10]. SORL1-deficient mice have higher levels of amyloid-β [1], the main component of the neuritic plaques (NPs) [11]. Similarly, a reduced SORL1 expression has been found in Alzheimer’s disease (AD) patients [12].

Recent studies indicate that single nucleotide polymorphism (SNP) in the *SORL1* gene is associated with LOAD [6,7,9,10,13,14,15,16]. Some of the polymorphic changes have been involved in amyloid formation and impairing the peptide physiological functions. While *SORL* variants were associated with LOAD in North European and Hispanic Caribbean family studies [10], no association was reported in a Caucasian American cohort [17]. The controversial results point out the relevance for considering ancestry admixture analysis in genetic association studies. In the present work, 12 SNPs in the SORL1 gene were genotyped in Mexican LOAD patients. Additionally, an interaction analysis of each SNP with the ancestry and with the presence of *APOE* ε4 allele was performed. The rs1784933 variant might be related to LOAD risk (A/A vs. G/G+A/G *p* = 0.03 OR = 1.608 (1.046–2.473)). Moreover, two linkage disequilibrium blocks were identified. Significant odds ratio values were observed in the logistic regression and MDR analysis between Block 1 and the rs1784933 polymorphism (OR = 5.539 (3.701–8.289) *p* = 0.0001) as well between Block 2 and the ApoE ε4 allele (OR = 30.334 (18.222–50.495), *p* = 0.0001).

## 2. Materials and Methods

### 2.1. Subjects

This study was carried out according with the ethical standards of the Committee on Human Experimentation of the institution (Instituto Nacional de Neurología y Neurocirugía Number 30/09). The experiments were done in accordance with the Helsinki Declaration of 1975. Participants were classified as possible late-onset Alzheimer’s (LOAD) or healthy subjects (control). Both groups were previously diagnosed by geriatricians, neurologists, and psychiatrists according to the National Institute of Neurological and Communicative Disorders and Stroke–Alzheimer’s Disease and Related Disorders Association (NINCDS-ADRDA) [18] criteria. The LOAD patients were interviewed at their scheduled visits at the Geriatric Clinic in the “Mocel” General Hospital in Mexico City, Instituto Nacional de Neurología y Neurocirugía, Hospital de Alta Especialidad de Ixtapaluca and the Geriatric Center in Querétaro. All participants (156 LOAD patients and 221 controls) signed an informed consent sheet. In the case of LOAD patients, additional consent of the primary care was required.

### 2.2. SNP Identification

Twelve polymorphisms in SORL1 were selected from the website www.alzforum.org (accessed on 9 September 2019) and the original study published by Rogaeva et al. [10] (Appendix A). The *APOE* genotype was determined as previously described by Forlenza using two SNPs (rs7412 and rs429358) [19]. Additionally, seven ancestry-informative markers (AIMs) were analyzed [20,21,22]. These ancestry markers had previously been used in populations of Latin American origin [20,21,22,23].

### 2.3. DNA Extraction and Genotyping

Peripheral blood was extracted from all subjects, and was stored in Vacutainer^®^ tubes with EDTA. Genomic DNA was isolated from whole blood by a QIAamp^®^ DNA Blood Midi Kit according to the manufacturer’s recommendations. Samples were stored at −20 °C until use. Genotypes were determined by allelic discrimination in a 7500 FAST Real-Time PCR System (Applied Biosystems, Waltham, CA, USA). The Real-Time PCR reactions were conducted according to the standard protocol, using allele-specific TaqMan probes (Applied Biosystems, Waltham, CA, USA).

### 2.4. Ancestry Analysis

Ancestry analysis was carried out genotyping 7 AIMs (rs4884, rs2695, rs17203, rs2862, rs3340, rs1800498 and rs2814778) in the LOAD and control samples [23]. AIMs frequency in Caucasian, African and Amerindian ancestral populations was obtained from the 1000 Genomes Project [24] and Salari et al. 2005 [21]. Admixture proportions in cases and controls were estimated in STRUCTURE software [25] and compared with a Fisher’s exact test.

### 2.5. SORL1 Polymorphisms and ApoEε4 Carriers

We conducted an interaction test between the genotype frequencies of 12 SNPs in SORL1 and the *ApoEε*4 carriers and *none4* carriers by Chi-square test with Epi Info.

### 2.6. Statistical Analysis

Categorical variables and genotype distribution were shown as numbers and percentages (%). The Hardy–Weinberg equilibrium (HWE) was evaluated using SNPstats (https://www.snpstats.net/start.htm, accessed on 9 September 2019) [26,27]. Disease associations were analyzed by logistic regression analysis adjusted by sex, age and ancestry.

The haplotype frequencies were determine in ARLEQUIN 3.11 software (University of Bern, Bern, Switzerland) and analyzed with a logistic regression model adjusted by age, gender and ancestry. Linkage disequilibrium (LD) among the selected SNPs was calculated using Haploview v.4.2 software [28].

Statistical analyses were performed by IBM SPSS (SPSS Inc., Chicago, IL, USA), and statistical significance was established at an α level of 0.05.

### 2.7. MDR Analysis

In order to study the epistasis, the multifactorial dimensionality reduction analysis (MDR) was assessed in the MDR v3.0.2 statistical package with Ritchie’s algorithm [29]. Multifactorial dimensionality reduction (MDR) is a statistical approach to detect and characterize combinations of attributes or independent variables that interact to influence a dependent or class variable. MDR was specifically designed to identify non-additive interactions between discrete variables influencing a binary outcome and is considered a non-parametric alternative, although the data was corroborated by a logistic regression model.

For our data, the MDR consists of two steps. First, the best multifactor combination is selected, and then the genotype combinations are classified into high- and low-risk groups for the models. Interaction analyses were performed using the open source MDR software package (MDR3.0.2) available at www.epistasis.org (https://github.com/EpistasisLab/scikit-mdr or https://ritchielab.org/software/mdr-download, accessed on 9 September 2019). This software allows us to visually analyze the interactions by means of a dendrogram and a Fruchterman–Rheingold graph, as well as the construction of the best models and values of accuracy, sensitivity, specificity and risk; these models have been applied in different diseases [30,31,32,33].

The best prediction model was selected by maximum testing and training balance accuracy (TBA) and cross-validation consistency (CVC). The model with the highest CVC, TBA and TrBA was tested by 1000 fold permutation testing and χ^2^ test at 0.05% significance levels during MDR analysis. The interaction entropy graphs were constructed based on MDR results to determine synergistic and non synergistic interactions among the variables [34]. The entropy graphs comprise nodes containing percentage entropy of each variable and connections joining them pairwise, showing entropy of interaction between them. Values inside nodes indicate information gain (IG) of individual attributes or main effects, whereas values between nodes show IG of pairwise combinations of attributes or interaction effects. Positive entropy (plotted in red or orange) indicates interaction, while negative entropy (plotted in green) indicates redundancy.

For the MDR model, age was coded as a dichotomous variable in 0 as <75 years and 1 as ≥75. The first analysis performed was on rs1784933 with Block 1 (rs668387, rs689021, rs641120) and the adjustment with age and sex, then the ApoE with Block 2 (rs3824968, rs1010159, rs1699102, rs2070045, rs2282649, rs3824968) and the adjustment for age, gender, and ancestry.

## 3. Results

### 3.1. Study Population

A total of 377 individuals were genotyped, 156 LOAD and 221 controls. The mean age onset for patients was 76.14 ± 8.8 years, and control subjects had a mean age of 73.64 ± 8.5 years (Table 1). While the age average between the groups is similar, results from the Mann–Whitney U test comparing the mean age between LOAD and controls showed significant differences (*p* = 0.008). For this reason, the genetic analysis was performed under a regression model considering at least the age as a variable.

### 3.2. Testing for Hardy–Weinberg Equilibrium

Hardy–Weinberg equilibrium tests were performed on LOAD and control groups in the twelve loci analyzed. The rs661057 and rs12285364 presented a significant Hardy–Weinberg equilibrium deviation (*p* < 0.05) and were not included in subsequent analyses.

### 3.3. Ancestry Analysis

The contribution of ancestral populations in each study group was calculated. Amerindian ancestry was represented in a higher proportion in control and LOAD, followed by Caucasian and finally in a lower proportion by African ancestry. We did not find significant differences (*p* = 0.66) in ancestry proportions between both groups. We concluded that both groups are genetically homogeneous in ancestry (Appendix A).

### 3.4. Analyzed SNPs and LOAD Risk

Table 2 summarizes the results from the SNP association study. Most of the SORL1 polymorphisms analyzed did not present differences in allelic or genotypic distributions between LOAD and controls. However, in the rs1784933, we identified significant differences in the distribution of genotypic frequency (A/A vs. G/G+A/G) in the AD group (*p* = 0.03, OR = 1.608 (1.046–2.473)). A modest association in the rs1010159 polymorphism was observed (A/A vs. G/G+A/G *p* = 0.05, OR = 1.590 (0.995–2.541)). *APOEε4* allele resulted more frequently in the LOAD group than in controls (*p* = 0.000, OR = 3.63 (2.195–6.004)). As can be seen in Appendix A, the rs1784933 (bold) polymorphisms have been related to the risk of LOAD in the Chinese population, similarly to the data found in the present study in Mexican population. They are not consistent with the previously studied populations. Similar results were observed in the rs2070045 polymorphism. Likewise, these can vary concerning the result obtained in our Mexican population sample depending on the population studied (Appendix A).

### 3.5. Haplotype Analysis

In our Mexican population samples, the LD mapping of the *SORL1* SNPs showed two main LD blocks (Figure 1).

The highest values of *D*’ were found between Block 1 (SNPs rs668387, rs689021 and rs641120 found), and Block 2 (SNPs rs2070045, rs3824966, rs1699102, rs3824968, rs2282649 and rs1010159). We were able to identify the risk haplotype TATGGCATGG presented in the LOAD group (*p* = 0.013, OR = 4.92 (1.40–17.28)). Additionally, the haplotype TATTCTTCAA could have a protective effect (*p* = 0.014, OR = 0.37 (0.17–0.81)) (Appendix A). Finally, we analyzed three risk haplotypes previously reported by Rogaeva et al.: CGC (8-9-10), CTT (22-23-24) and TTC (23-24-25). However, the haplotype distribution in our groups was no different (Appendix A).

### 3.6. Evaluation of Gene–Gene Interactions: MDR

To identify epistatic interaction candidates related to LOAD susceptibility, we applied Multifactorial Dimensionality Reduction Analysis (MDR) to detect which polymorphisms could be important for the prediction of the disease.

Suggestive interactions are shown in Table 3. The most important finding was identifying the best model to predict the susceptibility of LOAD by the interaction of the SNPs rs1784933 with Block 1 (rs668387, rs689021, rs641120) (OR = 3.097, 95% CI: 1.750–5.492). The interaction values increased when adjusted for age, gender and ancestry (OR = 5.539, 95% CI: 3.701–8.289). In the same way, a highly significant interaction was observed between ApoEε4 allele and Block 2 (rs2070045, rs3824966, rs1699102, rs3824968, rs2282649 and rs1010159) (OR = 3.372, 95% CI 2.007–5.665). Moreover, the interaction increased when adjusted for age, gender and ancestry (OR = 30.334, 95% CI 18.222–50.495). This model had the highest accuracy of 59.4%, 76.82% precision and consistency of 10/10.

In Figure 2, the combinations of significant low- and high-risk SNPs are shown, as well as an interaction graph based on entropy. The results are presented for each block found. Analysis of interaction dendrogram provided by MDR confirmed the implication of these factors in epistatic effects, indicating strong positive (synergistic) interactions between rs1784933 and rs689021 (dotted box). In the same way, a positive interaction was found between the polymorphism rs1784933 with rs668387 and rs641120 (Figure 2, Block 1A). These SNPs’ relations are associated with the susceptibility to dementia between cases and controls in our population. The Interaction graph confirmed the significance of the SNP interaction (Figure 2, Block 1B). In Block 2, the interaction dendrogram shows a strong positive interaction between rs2070045 and rs3824966. The entropy-based interaction graph confirmed strong positive effects between rs2070045 and rs3824966.

## 4. Discussion

The relation between SORL1 polymorphisms and APOE e4 allele has been studied mostly in Caucasian, Asiatic and Hispanic Caribbean cohorts, failing to reproduce the genetic association of the variants among these populations (Appendix A) [13,35,36,37,38,39]. Genetic admixture of samples could generate a stratification effect linked with false or positive associations. In the Mexican population, 93% is constituted by Mestizos, a complex biological admixture between Native Americans, Spaniards and African slaves [40]. For this reason, it is crucial to validate genetic variants associated with the risk of developing LOAD in our population.

In the present study, we evaluated SORL1 polymorphisms mostly located in intronic regulatory regions, which may modulate the expression of sortilin protein and thereby differently affect the risk of developing the disease [41,42]. To avoid the effects of stratification, the ancestry proportion in each group was estimated and performed with age and gender information under a regression model [23,43].

The rs2070045 polymorphism has been widely reported on in the literature and is located in the coding region of A-repeats of the LDLR-like domain, involved in cholesterol metabolism. Several studies have shown its strong association with the development of Alzheimer’s disease. Xue et al. reported that the T-allele of the rs2070045 polymorphism exerted a strong influence on the development of Alzheimer’s disease in a Chinese population [44]. Other authors have pointed out an association between rs2070045-G allele and increased CSF-tau with more hippocampal atrophy, both markers of neuronal injury and neurodegeneration [45]. Caglayan et al. showed that the rs2070045-G allele was associated with low expression of the sortilin receptor in brain tissue from confirmed AD patients, suggesting that insufficient receptor activity in the brain is an important risk factor in AD [46]. The reduced expression of SORL1 has been associated with increased Aβ peptide production [10].

APOE remains the major genetic risk factor of LOAD, increasing the probability 2–3-fold in individuals with one copy of APOE ε4 [7,9,16,47,48]. APOE ε4 is involved in the abnormal cleavage of the AβPP. Moreover, APOE ε4 modulates the cellular uptake of Aβ and could be related to SORL1 expression and its activity [48,49]. APOE ε4 carriers have a higher concentration of SORL1 protein in the cerebrospinal fluid but lower concentrations in some brain regions, such as hippocampus [41,48]. Other authors have suggested that the presence of a copy of the ε4 allele may have an effect on the genotypes of the various polymorphisms and have a cumulative effect on the development of the disease [7,44,50]. Several previous studies have reported SORL1/APOE interactions on the risk for AD. For this reason, we conducted an interaction test analysis of SORL1 SNPs in APOE 4 carriers.

The results showed an additive interaction between SORL1 and APOE in the studied population. We found a significant association between APOE ε4 carriers and the variants that form Block 2, particularly with the rs2070045 polymorphism. Similar reports indicate an altered hippocampal rsFC in carriers with risk of APOE ε4 or SORL1 G-allele, which may predispose these risk-allele carriers to be susceptible to development of AD [51]. We also observed a narrow relationship between the different blocks analyzed and the gender, suggesting the importance of integrating sex and genetic susceptibility. In the same way, Liang et al. observed men with the G/G genotype presented reduced integrity of the left cingulum hippocampal compared with G/T men. In contrast, women with the T/T genotype exhibited reduced integrity compared with G/G women, indicating a sex-moderated association of the SORL1 rs2070045 polymorphism and executive function [52].

It has been proposed that the effect of SORL1 genetic variants on AD risk could be specific for ethnic groups; if so, the discrepancy observed in different populations could be explained [13,35,36,37,38,39]. Based on the assumption previously discussed about genetic admixture of our population, we perform a multiple regression analysis considering SORL1 genotype frequencies within ancestry proportions.

In general, there is a consistent pattern in the results described in our present work, with major previous literature reports. The main finding obtained in this study indicates that SORL1 has a strong influence on the development of LOAD in our population. The relevance of the two blocks identified in the present work is supported by previous studies in European populations [10,53]. Our findings indicate a direct association of the rs1784933 polymorphism with the development of Alzheimer’s disease in the Mexican population. We also found a strong interaction of this polymorphism with the variants present in Block 1 (rs668387, rs689021 and rs641120), suggesting a hidden relationship between these variants (Haplotype TAT) and the development of Alzheimer’s in our population, which could be explained by high allelic heterogeneity. It is essential to point out the strong interaction between APOE ε4 carriers and the polymorphisms in Block 2 (G-rs2070045, G-rs3824966, C-rs1699102, A-rs3824968, T-rs2282649 and G-rs1010159).

In particular, the results suggest that the G-rs2070045 and G-rs3824966 polymorphisms could be used as risk markers due to their high interaction values, which increase significantly when adjusted for age, gender and ancestry. The high values of LD found in these two variants could suggest the existence of a high allelic heterogeneity that would justify a broader investigation of other underlying genetic variants.

The sample size of this study is seen as a limitation; however, there are no reports in the literature of this analyzed locus in Mexico. This work can approximate the genetic distribution of these markers in the Mexican population. Our results can support an interaction of several variants as one of the genetic causes for LOAD. In particular, we observe that the effects of the genotypes’ risk present in the different polymorphisms of SORL are also dependent on the presence of the APOE ε4 allele.

## 5. Conclusions

Our results could confirm previous reports that several SORL1 genetic variants are associated with LOAD, and the risk can be increased by the presence of the APOE e4 allele. We believe it is of great importance to increase the sample size to determine whether the observed discrepancies between previous reports are caused by a confounding factor or are directly related to the risk of pathology. Finally, this may be a preliminary study to determine the distribution of the genetic SORL1 markers associated with LOAD risk in the Mexican population.

## Figures and Tables

**Figure 1 genes-13-00587-f001:**
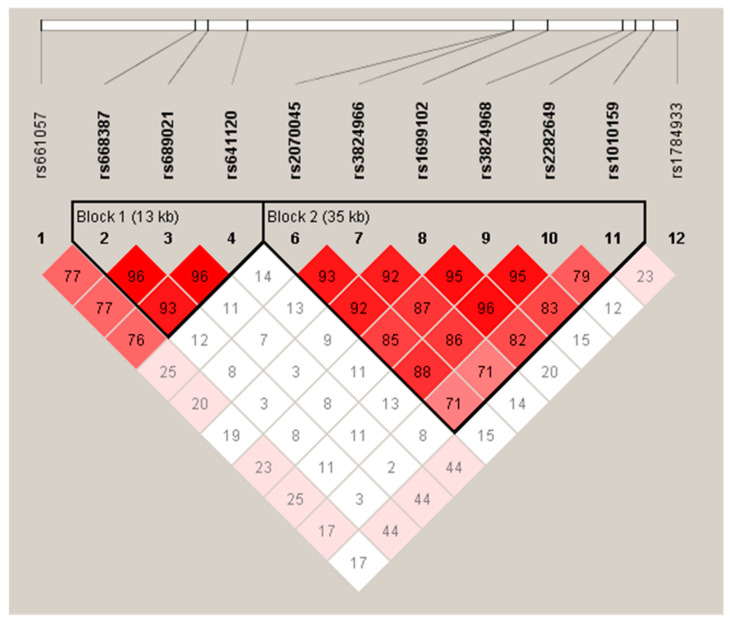
Linkage disequilibrium (LD) of *SORL1* SNPs in Mexican Population. Genotypes of twelve SNPs from the total samples (*n* = 377) were used to determine LD using Haploview software. *D’* values are shown within cells and standard LD color scheme was used, with white to red colors representing the increasing strength of LD. The highest values of *D’* were found between the SNPs rs668387, rs689021 and rs641120 (Block 1), and between the SNPs rs2070045, rs3824966, rs1699102, rs3824968, rs2282649 and rs1010159 (Block 2).

**Figure 2 genes-13-00587-f002:**
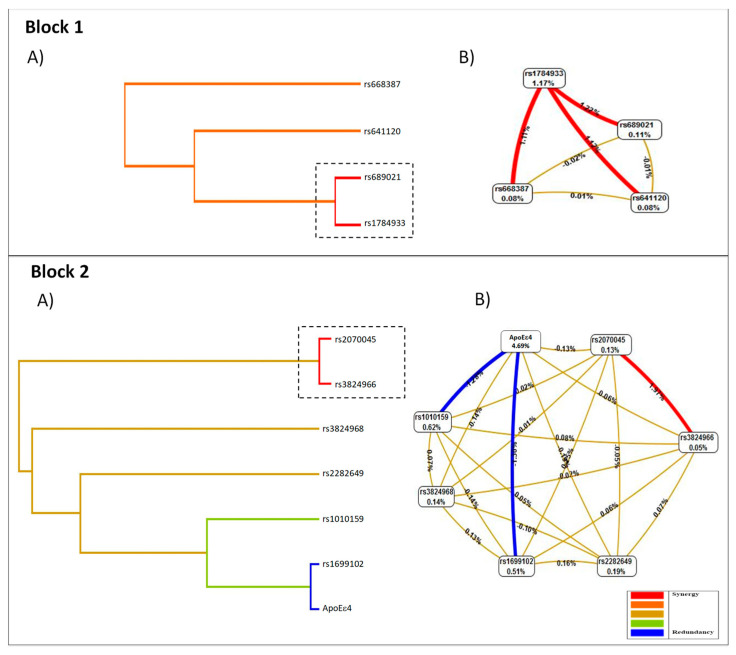
Interaction dendrogram and entropy-based interaction. Block 1 (**A**), the analysis interaction dendrogram, indicates a synergistic interaction between rs1784933 and rs689021 (dotted box) and to a lesser extent between the polymorphism rs1784933 with rs668387 and rs641120; Block 1 (**B**) entropy-based interaction graph shows the positive effects between these polymorphisms. Block 2 (**A**) the interaction dendrogram indicates a strong positive interaction between rs2070045 and rs3824966 (dotted box). (**B**) The entropy-based interaction graph confirmed strong positive effects between both polymorphisms. The models are adjusted for gender and age.

**Table 1 genes-13-00587-t001:** Demographic characteristics of subjects.

Study Group	Number of Women	Age Women (Years) (Mean ± SD)	Number of Men	Age Men (Years) (Mean ± SD)	Total	Total Mean Age	* p * Value
LOAD	106 (67.9%)	76.24 ± 8.59	50 (32.1%)	73.22 ± 9.56	156	76.14 ± 8.8	0.008 ^a^
Controls	167 (75.6%)	73.01 ± 8.7	54 (24.4%)	75.6 ± 7.298	221	73.64 ± 8.5	

^a^ Mann–Whitney U. Abbreviations: LOAD, Late Onset Alzheimer’s Disease; SD, standard deviation.

**Table 2 genes-13-00587-t002:** Genotype distribution of SORL1 variants with Alzheimer’s disease in Mexican samples.

Polymorphisms		Genotype Frequency (%)			HWE *p*	Model Inheritance		
**rs668387**		**C/C**	**T/T**	**T/C**			*p*	OR (95%)
	Controls (*n* = 221)	68 (30.8)	39 (17.6)	114 (51.6)	0.459	G/G vs. A/A+A/G	0.976	1.007 (0.640–1.584)
	Cases (*n* = 156)	47 (30.1)	24 (15.4)	85 (54.5)	0.155	G/G+A/G vs. A/A	0.758	1.093 (0.621–1.923)
						G/A vs. G/G+A/A	0.84	1.044 (0.687–1.587)
**rs689021**		**G/G**	**A/A**	**G/A**				
	Controls (*n* = 221)	66 (29.9)	39 (17.6)	116 (52.5)	0.329	C/C vs. T/T+C/T	0.713	1.089 (0.692–1.714)
	Cases (*n* = 156)	48 (30.8)	23 (14.7)	85 (54.5)	0.139	C/C+C/T vs. T/T	0.62	1.155 (0.653–2.045)
						C/T vs. C/C+T/T	0.978	1.006 (0.662–1.530)
**rs641120**		**C/C**	**T/T**	**C/T**				
	Controls (*n* = 221)	68 (30.8)	38 (17.2)	115 (52.0)	0.37	C/C vs. T/T+C/T	0.743	1.078 (0.687–1.692)
	Cases (*n* = 156)	49 (31.4)	23 (14.7)	84 (53.8)	0.179	C/C+C/T vs. T/T	0.657	1.139 (0.642–2.019)
						C/T vs. C/C+T/T	0.981	1.005 (0.661–1.527)
**rs2070045**		**G/G**	**T/T**	**G/T**				
	Controls (*n* = 221)	68 (30.8)	50 (22.6)	103 (46.6)	0.359	T/T vs. G/G+G/T	0.766	1.078 (0.659–1.762)
	Cases (*n* = 156)	42 (26.9)	38 (24.4)	76 (48.7)	0.755	T/T+G/T vs. G/G	0.469	1.186 (0.748–1.880)
						G/T vs. G/G+T/T	0.684	1.090 (0.719–1.653)
**rs3824966**		**C/C**	**G/G**	**C/G**				
	Controls (*n* = 221)	54 (24.4)	65 (29.4)	102 (46.2)	0.267	C/C vs. G/G+C/G	0.869	1.041 (0.644–1.684)
	Cases (*n* = 156)	40 (25.6)	42 (26.9)	74 (47.4)	0.523	C/C+C/G vs. G/G	0.635	1.119 (0.704–1.778)
						C/G vs. C/C+G/G	0.775	1.063 (0.701–1.612)
**rs1699102**		**C/C**	**T/T**	**C/T**				
	Controls (*n* = 221)	103(46.6)	26 (11.8)	92 (41.6)	0.436	T/T vs. C/C+C/T	0.221	1.454 (0.799–2.647)
	Cases (*n* = 156)	62 (39.7)	26 (16.7)	68 (43.6)	0.323	T/T+C/T vs. C/C	0.181	1.334 (0.874–2.036)
						C/T vs. T/T+C/C	0.626	1.110 (0.729–1.690)
**rs3824968**		**A/A**	**T/T**	**A/T**				
	Controls (*n* = 221)	65 (29.4)	47 (21.3)	109 (49.3)	0.917	T/T vs. A/A+A/T	0.497	1.187 (0.724–1.946)
	Cases (*n* = 156)	41 (26.3)	38 (24.4)	77 (49.3)	0.876	T/T+A/T vs. A/A	0.626	1.123 (0.704–1.791)
						A/T vs. T/T+A/A	0.895	0.972 (0.642–1.474)
**rs2282649**		**C/C**	**T/T**	**C/T**				
	Controls (*n* = 221)	52 (23.5)	65 (29.4)	104 (47.1)	0.409	C/C vs. T/T+C/T	0.695	1.101 (0.680–1.783)
	Cases (*n* = 156)	41 (26.3)	39 (25.0)	76 (48.7)	0.75	C/C+C/T vs. T/T	0.482	1.184 (0.739–1.896)
						C/T vs. C/C+T/T	0.773	1.063 (0.702–1.611)
**rs1010159**		**G/G**	**A/A**	**A/G**				
	Controls (*n* = 221)	71 (32.1)	50 (22.6)	100 (45.2)	0.197	A/A vs. G/G+A/G	0.053	1.590 (0.995–2.541)
	Cases (*n* = 156)	43 (27.6)	49 (31.4)	64 (41.0)	0.026	A/A+A/G vs. G/G	0.359	1.239 (0.784–1.956)
						A/G vs. A/A+G/G	0.381	0.829 (0.544–1.262)
**rs1784933**		**A/A**	**G/G**	**A/G**				
	Controls (*n* = 221)	71 (32.1)	50 (22.6)	100 (45.2)	0.197	A/A vs. G/G+A/G	0.03	1.608 (1.046–2.473)
	Cases (*n* = 156)	68 (43.6)	24 (15.4)	64 (41.0)	0.175	A/A+A/G vs. G/G	0.123	1.538 (0.890–2.656)
						A/G vs. A/A+G/G	0.38	0.828 (0.544–1.261)
**APOE**		**APOEε4 non carriers**		**APOEε4 carriers**				
	Controls (*n* = 221)	188 (85.1)		33 (14.9)		ε4 carriers vs. Non carriers	0.000	3.630 (2.195–6.004)
	Cases (*n =* 156)	98 (62.8)		58 (37.2)				

Logistic regression adjusting for sex and age.

**Table 3 genes-13-00587-t003:** Risk assessment according to Haplotypes (Block 1 and 2) using the associated SNPs (rs1784933 and ApoEε4).

Model	Sensitivity	Specificity	Precision	OR (95% CI)	*p* Value
	(%)	(%)	(%)		
Block 1	0.456	0.609	0.451	1.304 (0.842–2.020)	0.2341
rs1784933	0.436	0.679	0.489	1.633 (1.044–2.553)	0.0311
Block 1 and rs1784933	0.868	0.32	0.474	3.097 (1.750–5.492)	0.0001
Block 1 and rs1784933 *	0.673	0.632	0.564	3.531(2.641–6.845)	0.0001
Block 1 and rs1784933 **	0.699	0.704	0.625	5.539 (3.701–8.289)	0.0001
Block 2	0.634	0.617	0.539	2.794 (1.787–4.368)	0.0001
ApoEε4	0.372	0.851	0.637	3.372 (2.007–5.665)	0.0001
Block 2 and ApoEε4	0.62	0.796	0.682	6.372 (3.924–10.347)	0.0001
Block 2 and ApoEε4 *	0.769	0.731	0.719	10.706 (6.783–16.899)	0.0001
Block 2 and ApoEε4 **	0.813	0.875	0.821	30.334 (18.222–50.495)	0.0001

Block 1 (rs668387, rs689021, rs641120), Block 2 (rs2070045, rs3824966, rs1699102, rs3824968, rs2282649, rs1010159) logistic regression and MDR analysis; *p* < 0.05. OR, odds ratio; CI, confidence interval. * Adjusted for age and gender. ** Adjusted for age, gender and ancestry.

## Data Availability

Not applicable.

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
