# Peer review of "SORL1 Polymorphisms in Mexican Patients with Alzheimer’s Disease"

_genes, 2022, doi:10.3390/genes13040587_

Round 1

Reviewer 1 Report

This manuscript studies the correlation of 12 different SORL1 SNPs with Late-Onset Alzheimer disease (LOAD) and identify two SNPs namely rs1784933 and rs1010159 to be associated with higher risk for LOAD. The authors further describe interaction of rs1010159 with other several other genetic markers. The authors also report interaction of SORL1 (the rs2070045 polymorphism) and APOE and suggests sexual dimorphism in the observed interaction. The findings of this study will be of interest to the readers of this journal. The manuscript is well written, and the methods are clearly and satisfactorily described. There are some minor points that the authors need to address:

  1. The age of the control and the test populations are significantly different. This might have an effect of the outcomes of the studies. The authors need to address this in the manuscript.
  2. The sentences 'This section...can be drawn' (Page 5, lines 22-25) looks out of place and should be deleted.

Author Response

Thank you very much for the reviewer's comments and observations. To avoid any effect, due to the difference found in the age between the different groups, a genetic analysis was performed under a regression model considering at least age as a variable. 

Changes made to the manuscript are marked in red.

Reviewer 2 Report

The manuscript by Rios and coworkers describes the analysis of sortilin related receptor (SORL1) gene polymorphisms in Mexican late-onset Alzheimer disease (AD) populations.

Overall, this is a short article and highlights an important aspect of the AD field. Overall, the conclusions drawn are generally substantiated by evidence, there are some points that need clarification. A series of questions remain open, and the article would profit from additional information or at least from a more focused discussion of these points:

-The paper has yet to be tightened in terms of its language. It requires shortening throughout to improve clarity. The abstract should succinctly state the findings in the patients identified.

-The discussion needs focus to highlight the important findings (the first paragraph of the discussion is superfluous).

-The method section is poorly described. It needs more substantial information.

-Can authors make a table for comparing different ethnic AD populations studies showing SORL1 SNPs. This would be important to substantiate their finding and interpretation.

Author Response

Thank you very much for the observations and comments made by the reviewer. Changes made to the manuscript are marked in red.

Point 1: The paper has yet to be tightened in terms of its language. It requires shortening throughout to improve clarity. The abstract should succinctly state the findings in the patients identified.

Response 1: The manuscript has been shortened for better clarity in content. Also, the abstract points out the main findings found in this study

Point 2: The discussion needs focus to highlight the important findings (the first paragraph of the discussion is superfluous).

Response 2: The discussion has been restructured and shortened for better clarity in content. The paragraph indicated by the reviewer has been eliminated

Point 3: The method section is poorly described. It needs more substantial information.

Response 3: The methodology was modified, and each of the points is specifically explained.

Point 4: Can authors make a table for comparing different ethnic AD populations studies showing SORL1 SNPs. This would be important to substantiate their finding and interpretation.

Response 4: Two tables that compares the allelic and genotypic frequencies between different populations and the findings found in the Mexican subjects, was made (Table 2S and 3S Supplementary Material)

Round 2

Reviewer 2 Report

The authors have now revised the manuscript to deal with my original concerns.

Author Response

The manuscript was revised to correct language mistakes.